# A prospective matched case-control study on the genomic epidemiology of colistin-resistant Enterobacterales from Dutch patients

Karuna E. W. Vendrik [1,2✉], Angela de Haan[1], Sandra Witteveen[1], Antoni P. A. Hendrickx [1✉], Fabian Landman[1], Daan W. Notermans[1,3], Paul Bijkerk[1], Annelot F. Schoffelen[1], Sabine C. de Greeff[1], Cornelia C. H. Wielders [1], Jelle J. Goeman[4], Ed J. Kuijper[1,2], Leo. M. Schouls[1] & ColRE survey consortium*

### Abstract

**Background** Colistin is a last-resort treatment option for infections with multidrug-resistant Gram-negative bacteria. However, colistin resistance is increasing.

**Methods** A six-month prospective matched case-control study was performed in which 22 Dutch laboratories with 32 associated hospitals participated. Laboratories were invited to send a maximum of five colistin-resistant *Escherichia coli* or *Klebsiella pneumoniae* (COLR-EK) isolates and five colistin-susceptible isolates (COLS-EK) to the reference laboratory, matched for patient location, material of origin and bacterial species. Epidemiological/clinical data were collected and included in the analysis. Characteristics of COLR-EK/COLS-EK isolates were compared using logistic regression with correction for variables used for matching. Forty-six ColR-EK/ColS-EK pairs were analysed by next-generation sequencing (NGS) for whole-genome multi-locus sequence typing and identification of resistance genes, including *mcr* genes. To identify chromosomal mutations potentially leading to colistin resistance, NGS reads were mapped against gene sequences of *pmrAB*, *phoPQ*, *mgrB* and *crrB*.

**Results** In total, 72 COLR-EK/COLS-EK pairs (75% *E. coli* and 25% *K. pneumoniae*) were included. Twenty-one percent of COLR-EK patients had received colistin, in contrast to 3% of COLS-EK patients (OR > 2.9). Of COLR-EK isolates, five contained *mcr-1* and two *mcr-9*. One isolate lost *mcr-9* after repeated sub-culturing, but retained colistin resistance. Among 46 sequenced COLR-EK isolates, genetic diversity was large and 19 (41.3%) isolates had chromosomal mutations potentially associated with colistin resistance.

**Conclusions** Colistin resistance is present but uncommon in the Netherlands and caused by the *mcr* gene in a minority of COLR-EK isolates. There is a need for surveillance of colistin resistance using appropriate susceptibility testing methods.

### Plain language summary

The antibiotic colistin is the last option for treatment of patients infected with bacteria that are not killed by other antibiotics (multi-drug resistant bacteria). However, the number of infections with bacteria that are also not killed by colistin (colistin-resistant bacteria) is increasing, hampering treatment of patients infected with these bacteria. We investigated the number and characteristics of patients carrying two types of colistin-resistant bacteria. Twenty-two Dutch laboratories provided bacteria from patients, of which some were colistin-resistant and of which some could be killed by colistin (colistin-susceptible). Patients with colistin-resistant bacteria had previously used colistin more frequently compared to patients with colistin-susceptible bacteria. In about 10% of patients with colistin-resistant bacteria, the inability of the bacteria to be killed by colistin was caused by a gene that could easily be transferred to other bacteria.

[1] Centre for Infectious Disease Control, National Institute for Public Health and the Environment (Rijksinstituut voor Volksgezondheid en Milieu, RIVM), Bilthoven, The Netherlands. [2] Department of Medical Microbiology, Leiden University Medical Center, Leiden, The Netherlands. [3] Department of Medical Microbiology, Amsterdam University Medical Centres, Location Academic Medical Centre, Amsterdam, The Netherlands. [4] Department of Biomedical Data Sciences, Leiden University Medical Center, Leiden, The Netherlands. *A list of authors and their affiliations appears at the end of the paper.
✉email: karunavendrik@gmail.com; antoni.hendrickx@rivm.nl

Multidrug-resistant Gram-negative bacteria are rapidly emerging worldwide[1–3]. The mean resistance percentage for carbapenem among *Klebsiella pneumoniae* isolates in Europe is 7.9%, with some countries reporting resistance percentages between 25 to 50% or ≥50%. It is observed in only 0.3% of *Escherichia coli* isolates[3]. The polymyxin colistin is a last resort treatment option against severe infections by multidrug resistant Gram-negative organisms (MDRO) and is increasingly used. However, colistin is potentially neuro- and nephrotoxic when administered parenterally.

Colistin has been used for decades for the prevention and treatment of infections caused by Enterobacterales in livestock[4,5]. In humans in the Netherlands, colistin is mainly used as part of the treatment of infections with *Pseudomonas aeruginosa* in nebulised form in patients with pulmonary diseases such as cystic fibrosis, as well as for topical treatment of otitis externa and ophthalmic infections[4,6]. In addition, colistin is used in prophylactic antibiotic regimens as a component of selective decontamination of the digestive tract (SDD) or selective oropharyngeal decontamination (SOD) with the aim to reduce infections and mortality in intensive care unit (ICU)-admitted patients and in neutropenic patients with a haematological disease[7]. Colistin is also used as last-resort treatment for MDRO.

Colistin resistance is increasing worldwide[8–10] and this poses problems in treatment of infections with MDRO. *K. pneumoniae* is the species most commonly involved in the development of colistin resistance[11]. Among 646 carbapenem-resistant *K. pneumoniae* found in Europe in 2013–2014, 28% had tested colistin-resistant[12]. Outbreaks of carbapenemase ($bla_{\text{NDM-1}}$ and $bla_{\text{OXA-48}}$)-producing and colistin-resistant *K. pneumoniae* have been reported in Europe and highlight the emerging threat that humans are currently facing[13]. The prevalence and incidence of colistin resistance is difficult to assess, as colistin susceptibility testing is usually not part of the initial routine testing panel for Enterobacterales and is methodologically challenging with several methods producing unreliable results. Broth microdilution is the gold standard method, but is labour-intensive and time-consuming. Methods such as disk diffusion and agar dilution produce unreliable results due to the large molecular size of colistin making it poorly diffusible through agars[14]. Furthermore, many laboratories use automated antimicrobial susceptibility testing (AST) systems with high very major error rates (producing false susceptible results)[15,16].

Several chromosomal mutations in bacteria can lead to colistin resistance. For *K. pneumoniae*, mutations in the chromosomally located *pmrAB, phoPQ, mgrB* and *crrB* genes have been intensively studied. Mutations in these genes lead to the upregulation of the modification of lipid A in lipopolysaccharide (LPS). This modification leads to decreased negative charge of the bacterial membrane impairing the interaction between colistin and LPS[17]. In *E. coli*, evidence on the role of chromosomal mutations in colistin resistance is scarce[18]. Colistin resistance in *E. coli* strains has been linked to *phoPQ* and *pmrAB* genes, but experimental validation is mostly lacking[18].

The risk for spread of colistin resistance is further increased by transferable plasmid-mediated colistin resistance (*mcr*) genes that can transmit colistin resistance more easily between bacteria, including bacteria from different species[19]. Until now, *mcr* genes 1 to 10 have been discovered. Notably, *E. coli* is the most abundant *mcr*-containing species[20,21]. A study that examined 457 *mcr-1*-positive Enterobacterales isolates from 31 different countries, found 411 *E. coli* isolates (89.9%)[20]. Colistin resistance by chromosomal mutations and *mcr* genes is mostly caused by adding cationic groups to LPS[22]. Colistin resistance may be triggered directly by selection during treatment with colistin[11] or indirectly during treatment with other antibiotics by co-transfer of the *mcr*

gene with other resistance genes on the same plasmid or different plasmids[23,24].

Little is known about the genomic epidemiology of colistin resistance in the Netherlands. One outbreak with six patients from a hospital and nursing home with a colistin-resistant carbapenemase-producing *K. pneumoniae* in 2013 has been described[25].

The objectives of this study were to determine the incidence and risk factors of patients colonised or infected with colistin-resistant *E. coli* or *K. pneumoniae* (COLR-EK) and to characterise the isolates. This study shows that colistin resistance is present but uncommon in the Netherlands and is plasmid-mediated in a minority of isolates.

## Methods
A prospective matched case-control study with density-based sampling was performed. This project took place between May 2019 and February 2020 and was part of a pan-European multicentre study on colistin- and carbapenem-resistant Enterobacterales (CCRE survey) of the European Centre for Disease Prevention and Control (ECDC)[26].

**Participating laboratories**. Twenty-two Dutch medical microbiology laboratories (MMLs) providing services for 32 hospitals, participated in this project using the infrastructure of a web-based laboratory network, called Type-Ned, which is used for the national carbapenemase-producing Enterobacterales surveillance in the Netherlands[27]. Laboratories were selected based on NUTS-2 regions (nomenclature of territorial units for statistics) of the associated hospitals. In the Netherlands, the provinces represent the NUTS-2 regions. At least one hospital site per NUTS-2 region had to be included[28]. Participating hospitals had to offer acute care services.

**Study population and isolates**. MMLs were requested to send a maximum of five COLR-EK isolates with a minimum inhibitory concentration (MIC) > 2 mg/L for colistin and/or a *mcr* gene that were collected in a 6-month period. In line with the European CCRE survey guidelines, only isolates not producing carbapenemases and with a meropenem MIC ≤ 0.25 mg/L were included. Controls were selected with density-based sampling: for each COLR-EK, the first following colistin-susceptible *E. coli* or *K. pneumoniae* (COLS-EK) with a colistin MIC ≤ 2 mg/L, no *mcr* gene and a meropenem MIC ≤ 0.25 mg/L matched with the COLR-EK for patient location (sender of the isolate: from community or hospital), patient material and bacterial species, was requested. Only a single isolate per patient was included in the study. MMLs were asked to send isolates which they classified as COLR-EK and COLS-EK based on their routine susceptibility testing.

**Detection of resistance genes and antimicrobial susceptibility testing**. Microbiological confirmation of all submitted isolates was performed at the Dutch National Institute for Public Health and the Environment (RIVM). Species assignments were confirmed by MALDI-TOF (Microflex LT System; Bruker, Leiderdorp, Netherlands). The absence of carbapenem resistance was assessed by the meropenem Etest (BioMérieux Inc., Marcy L'Étoile, France). The Carbapenem Inactivation Method (CIM)[29] was used to determine whether phenotypical carbapenemase production was absent. An in-house developed multiplex PCR assay was used to assess the presence of carbapenemase-encoding genes using primers that target the allelic variants of the following genes: $bla_{\text{NDM}}$, $bla_{\text{KPC}}$, $bla_{\text{VIM}}$, $bla_{\text{IMP}}$, and $bla_{\text{OXA-48}}$. The presence of *mcr* genes was assessed by using two specific in-house

multiplex PCRs for *mcr-1* to *mcr-5* genes and for *mcr-6* to *mcr-8* genes. Colistin resistance was confirmed using a standardised broth microdilution (BMD; Micronaut MIC strip colistin, Merlin) using an ECDC-recommended protocol[30] including one positive control (NCTC 13846) and three negative controls (ATCC 25922, ATCC 27853 and ATCC 700603). AST of colistin and meropenem was performed according to EUCAST detection guidance and breakpoints[31,32].

COLR-EK and COLS-EK isolates were classified as MDRO or non-MDRO, based on antibiograms received by participating laboratories (mostly results of VITEK automated testing). MDRO was defined according to definitions of the Dutch Working Group on Infection Prevention[33]: *E. coli* or *K. pneumoniae* that are extended-spectrum beta-lactamase (ESBL)-producing, that are resistant to both a fluoroquinolone and an aminoglycoside or that produce carbapenemases. ESBL-production was defined as resistance to ceftazidime and/or either cefotaxime or ceftriaxone with additional resistance to cefepime or susceptibility to cefoxitin. When results of Etest ESBL strips or combination disc methods[34] were available, these were used to define ESBL-producers.

**Genomic analysis.** Forty-six COLR-EK/COLS-EK pairs were subjected to next-generation sequencing (NGS) to assess genetic relatedness of strains, presence of chromosomal mutations leading to colistin resistance, antimicrobial resistance (AMR) genes, serotypes, plasmid replicons and virulence factors. The selection of isolates for NGS was based on a minimal time between sample dates of a colistin-resistant and the matched colistin-susceptible isolate, a sufficient number of both *E. coli* and *K. pneumoniae* isolates and a diverse selection of geographic locations. Isolates were subjected to NGS using the Illumina HiSeq 2500 (Base-Clear). Genetic relatedness was assessed with NGS data by classical and whole-genome multilocus sequence typing (wgMLST). A minimum spanning tree was created in Bionumerics version 7.6.3 (Applied Maths, Sint-Martens-Latem, Belgium) using an in-house wgMLST scheme in SeqSphere software version 6.0.2 (Ridom GmbH, Münster, Germany). The categorical coefficient was used to calculate the MST. For *K. pneumoniae*, this in-house wgMLST scheme was comprised of 4978 genes (3471 core-genome and 1507 accessory-genome targets) using *K. pneumoniae* MGH 78,578 (NC_009648.1) as a reference genome. For *E. coli*, 4503 genes (3199 core-genome and 1304 accessory-genome targets) were used with *E. coli 536* (CP000247.1) as a reference genome[27]. Genetic clusters were defined as collections of isolates with a maximum of 25 alleles differences for *E coli* and 20 for *K. pneumoniae*. For classical MLST, the existing schemes available via SeqSphere were used. AMR genes were identified via ResFinder software[35] and only AMR genes with ≥97% sequence identity with the reference sequences were included. The presence of *mcr* genes 1 to 10 was also analysed using BLAST (CLC Genomics Workbench version 20.0.3; Qiagen Bioinformatics, Aarhus, Denmark). The presence of plasmid replicons was assessed using PlasmidFinder software[36], including only replicons with 100% sequence identity and that were completely present. For the identification of serotypes and virulence factors, VirulenceFinder[37], SerotypeFinder[38] and Kleborate (https://github.com/katholt/Kleborate) were used. All NGS-derived data were imported into BioNumerics for subsequent analyses. Raw NGS sequence data of all sequenced isolates were deposited in the Sequence Read Archive and plasmids with *mcr* genes in GenBank of the National Centre for Biotechnology Information (NCBI) under BioProject ID PRJNA754858.

Isolates carrying *mcr* genes were subjected to third-generation sequencing (TGS; Oxford Nanopore, Oxford, United Kingdom).

For TGS, an in-house protocol was used to isolate high-molecular-weight DNA[39]. Bacteria were grown overnight in 1.5 ml Brain heart infusion broth. Subsequently, the culture was spun down for 2 min at $13,000 \times g$. We washed the pellet and resuspended it in 500 µl of NaCl (150 mM). The resulting suspension was spun down for 5 min at $5000 \times g$. We resuspended the pellet in 100 µl QuickExtract DNA Extraction Solution (Lucigen) and 0.1 µl Ready-Lyse Lysozyme solution (Epicentre). This was incubated at 37 °C for 1 h and then we added 85 µl 10 mM Tris 1 mM EDTA pH = 8 ($1 \times$ TE), 10 µl proteinase K (>600 mAU/mL, Qiagen) and 5 µl 20% sodium dodecyl sulfate solution. The mixture was incubated for 30 min at 56 °C. Subsequently, $0.1 \times$ volume 3 M sodium acetate pH = 5.2 and $2.5 \times$ volume ice-cold 100% ethanol were added and DNA was allowed to precipitate overnight at −20 °C. This was spun down for 15 min at $13,000 \times g$. The resulting pellets were washed with 1 ml 70% ethanol and this was spun down again for 5 min at $13,000 \times g$. The pellet was dried, dissolved in 200 µl $1 \times$ TE and diluted with Nuclease-free water to 1 µg.

After DNA isolation, we used the Oxford Nanopore protocol SQK-LSK109 (https://community.nanoporetech.com) and the expansion kit for native barcoding EXP-NBD104 (Oxford Nanopore Technologies)[39]. No optional shearing of DNA was performed. FFPE and end-repair kits (New England BioLabs) were used to repair the DNA. Barcodes were ligated with 1× bead clean up using AMPure XP (Beckman Coulter Nederland) after each step. Sequencing adaptors were added to pooled barcoded isolates by ligation. The final library was loaded onto a GridION flow cell (MIN-106 R9.4.1). Subsequently, GridION was used for live base calling using the MinKNOW GUI and afterwards de-multiplexing with the Guppy algorithm (ONT Guppy barcoding software version 3.5.1) on a Red Hat Enterprise Linux Server. Reads with length <5000 base pairs were omitted using NanoFilt (version 2.2.0), and then, 50 base pairs of both sides were trimmed using headcrop and tailcrop settings. In addition, FiltLong (version 0.2.0) was used to filter the reads with the 90% highest Qscore and make a subset up to a maximum of 500 Mb. The resulting FASTQ was used as input for assembly. The NGS and TGS data were used in Unicycler v0.4.8 for hybrid assembly[40] to reconstruct chromosomes and plasmids, which were annotated by Prokka v1.14.6 and loaded into BioNumerics for further analyses[41]. NanoPlot (v1.31.0) was used for quality control of the TGS data, where most isolates had coverages of at least 80x chromosome size with only three isolates 10–20x coverage. The read length N50 range was between 35 and 55 kb. NGS data were not trimmed before running Unicycler. Unicycler was run with default settings and verbosity 2. The default depth_filter setting of 0.25 was used. However, when the *mcr* gene that was found in the NGS data could not be retrieved in the TGS data, a depth_filter of 0.1 was used. Only contigs of >2.5 kb were analysed in this study. Plasmids containing *mcr* genes were compared with each other using chromosome comparison in BioNumerics and with other previously found *mcr*-containing plasmids in NCBI BLAST.

Isolates with the *mcr-9* gene were repeatedly subcultured, with the aim to cure the isolates from the *mcr-9* gene. The absence of the *mcr-9* gene was examined using PCR. In case absent, BMD was performed simultaneously on the cured and non-cured isolate and both isolates were subjected to NGS and TGS.

To identify mutations that may potentially lead to colistin resistance, NGS reads were mapped in CLC Genomics Workbench version 20.0.3 against the gene sequences of *pmrAB*, *phoPQ*, *mgrB* and *crrB*, present in the NCBI sequence database (references for *K. pneumoniae*: *crrB*—KY587106, *mgrB*—MN187248, *phoQ*—KY587110, *pmrA*—MG243721, *pmrB*—KJ626267, *phoP*—KY587067; *E. coli*: *phoP*—NZ_CP038353-Eco,

*phoQ*—NZ_CP038353-Eco, *pmrA*—NZ_CP038353-Eco, *pmrB*—NZ_CP038353 Eco).

**Metadata collection**. Clinical and epidemiological data were extracted from the electronic patient medical records by the participating laboratories and entered into Type-Ned. Microbiological data (including no of detected, tested and positive *K. pneumoniae* and *E. coli* isolates and the test on which submission of isolates was based), extracted from the local laboratory information system, and general hospital data (denominator data and use of colistin in SDD/SOD) were entered into web-based questionnaires. In addition, a questionnaire on local laboratory testing policies was composed in collaboration with the Dutch Infectious Disease Surveillance Information System-Antibiotic Resistance (ISIS-AR)[42] prior to start of this study and these data were extracted from ISIS-AR for participating laboratories after this study[42].

The minimal rate of carbapenem-susceptible COLR-EK isolates was calculated as the number of colistin-resistant *E. coli* and *K. pneumoniae* isolates, confirmed for this study, divided by the number of patient days or person years, provided by the participating laboratories (cases of hospitals without provided denominator data were subtracted).

**Ethical permissions and privacy**. The medical ethical committee of the University Medical Centre Utrecht has defined this study (19/262) as not falling under the scope of the Dutch law 'Wet medisch-wetenschappelijk onderzoek met mensen' ("Niet WMO-plichtig"). This means that no further medical ethical evaluation by the committee is needed since no additional individual patient data or isolates/materials were collected specifically for this study and no actions were requested from patients. Furthermore, the data collected in this study do not include personally identifiable information. Written or verbal informed consent was therefore not required. The collection and storage of data complied to the General Data Protection Regulation (EU 2016/679).

**Statistics and reproducibility**. Data are presented as *n* (%) for categorical variables and mean (standard deviation) or, for variables that have a skewed distribution, median and interquartile range (IQR) [first quartile-third quartile] for numerical variables. Categorical variables with characteristics of matched COLR-EK and COLS-EK isolates/patients were compared using matching-adjusted logistic regression (LR) with correction for variables used for matching (patient location, material of origin and bacterial species) and the odds ratio (OR) with 95% confidence interval (CI) were calculated. When necessary, a matching- and confounding-adjusted logistic regression was performed with additional covariates. For numerical variables with non-normal distribution, the Wilcoxon signed-rank test was used. For the comparison of virulence factors between colistin-resistant and -susceptible *E. coli*, data was corrected with a Bonferroni multiple testing correction. A two-sided *p*-value of <0.05 was considered statistically significant. No data were excluded from the analyses. In all analyses, a complete case analysis was performed. The number of patients with available data per variable are mentioned. Our hypothesis is that missing data were mostly missing completely at random, due to the substantial workload or difficulties to find certain information in the electronic patient files (most missing data were observed in variables, such as antibiotic use or colistin use in the previous 6 months and a profession with direct patientcare). However, persons that filled in the questionnaire were not blinded for colistin susceptibility testing results and therefore missing not at random cannot be ruled out. The researcher that analysed the data was also not blinded for colistin-susceptibility testing results. Randomization was not applicable. This study is set up with strict inclusion criteria and a highly structured protocol. Experimental data were generated using well established, reproducible, robust and validated methodology. Where applicable, positive and negative controls were included, which showed highly reproducible results. STATA SE version 15.1 (StataCorp, College Station, TX, USA) was used for data-analysis.

**Reporting summary**. Further information on research design is available in the Nature Research Reporting Summary linked to this article.

## Results
Twenty-two Dutch laboratories, providing services for 32 hospitals and/or primary care, participated in this project, covering all but two NUTS-2 regions in the Netherlands (NL23 and NL21). Colistin resistance was found in nine of ten participating NUTS-2 regions.

**Characteristics of patients**. Of 22 participating laboratories, 17 found COLR-EK isolates that met the inclusion criteria for this study. COLR-EK and COLS-EK isolates obtained from 72 patients per group were confirmed at the RIVM and were included in this study. This implied a median of 3 (IQR 2–6) patients per laboratory. Some laboratories sent more than the maximum of five COLR-EK isolates and these were also included when inclusion criteria were met. The most important patient characteristics are shown in Table 1 and a more detailed table with results of the logistic regression analysis in Supplementary Data 1. Most isolates derived from urine samples (*n* = 55 per group, 76.4%). Furthermore, 42 (58.3%) isolates were collected in hospitals, 24 (33.3%) in general practices and 6 (8.3%) in other healthcare facilities. Of 43 COLR-EK patients with available information on colistin use, 9 (20.9%) had received colistin in the previous 6 months, significantly more than 40 COLS-EK patients in which only a single patient (2.5%) had received colistin, resulting in a matching-adjusted OR of 58.3 (95% CI 2.9–1158.7; LR *p* = 0.008). Of those ten patients who had used colistin recently, two developed an infection caused by COLR-EK, located in the urinary tract. Colistin was used for SDD/SOD in all ten cases. Among 26 hospitals that provided information on colistin use in SDD/SOD, 20 (76.9%) hospitals provided SDD/SOD medication with colistin. Use of other antibiotics in the previous 6 months was observed in 73.2% of COLR-EK and 55.9% of COLS-EK patients (OR 2.8; 95% CI 0.9–8.8; LR *p* = 0.084). In the COLR-EK group, 22.2% had a malignancy, which was significantly higher than in the COLS-EK group with 10.3% (OR 4.7; 95% CI 1.1–19.6; LR *p* = 0.033). However, the effect disappeared after correction for colistin use (OR 2.6; 95% CI 0.4–15.8; LR *p* = 0.299). Four of 14 (28.6%) patients with known malignancies and data on colistin use had used colistin compared to none among 38 patients without malignancies.

**Testing for colistin resistance**. Among the 22 participating laboratories, colistin susceptibility testing policies differed substantially. Nine laboratories tested all Enterobacterales for colistin resistance. Seven laboratories tested for colistin resistance only for special reasons, i.e. when colistin was considered as treatment option (*n* = 2) or in case of a combination of factors, such as specific bacterial species and certain patient groups (*n* = 5). One laboratory never tested for colistin resistance except for this study. No information on the testing policies was available for the five remaining laboratories. Table 2 depicts the tests in use and the indications.

**Table 1 Characteristics of patients carrying COLR-EK or COLS-EK isolates.**

| | COLR-EK (N = 72) | | | COLS-EK (N = 72) | | |
|---|---|---|---|---|---|---|
| | N | N total | % | N | N total | % |
| **Median age (interquartile range)** | | 72 | 73.5 (IQR 56.0-83.0) | | 72 | 72.0 (IQR 51.5-78.0) |
| **Female** | 53 | 72 | 73.6% | 53 | 72 | 73.6% |
| **Species** | | | | | | |
| E. coli | 54 | 72 | 75.0% | 54 | 72 | 75.0% |
| K. pneumoniae | 18 | 72 | 25.0% | 18 | 72 | 25.0% |
| **Material** | | | | | | |
| Urine | 51 | 72 | 70.8% | 52 | 72 | 72.2% |
| Urine in case of bladder catheter | 4 | 72 | 5.6% | 3 | 72 | 4.2% |
| Rectum/perineum swab | 10 | 72 | 13.9% | 10 | 72 | 13.9% |
| Faeces | 3 | 72 | 4.2% | 3 | 72 | 4.2% |
| Wound secretion | 1 | 72 | 1.4% | 1 | 72 | 1.4% |
| Blood | 2 | 72 | 2.8% | 2 | 72 | 2.8% |
| Throat swab | 1 | 72 | 1.4% | 1 | 72 | 1.4% |
| **Sender of isolate** | | | | | | |
| Hospital | 42 | 72 | 58.3% | 42 | 72 | 58.3% |
| Inpatient | 27 | 42 | 64.3% | 36 | 42 | 85.7% |
| Outpatient | 15 | 42 | 35.7% | 6 | 42 | 14.3% |
| General practitioner | 24 | 72 | 33.3% | 24 | 72 | 33.3% |
| Nursing home/Elderly home/Care centre | 6 | 72 | 8.3% | 6 | 72 | 8.3% |
| **Infection** | 54 | 69 | 78.3% | 51 | 69 | 73.9% |
| **Comorbidity** | | | | | | |
| Renal insufficiency | 2 | 45 | 4.4% | 3 | 58 | 5.2% |
| Immunosuppression | 3 | 45 | 6.7% | 5 | 58 | 8.6% |
| Type 2 diabetes mellitus | 3 | 45 | 6.7% | 5 | 58 | 8.6% |
| Chronic obstructive pulmonary disease | 3 | 45 | 6.7% | 1 | 58 | 1.7% |
| Malignancy | 10 | 45 | 22.2% | 6 | 58 | 10.3% |
| Comorbidity | | | | | | |
| **Previous antibiotic use** | | | | | | |
| Use of colistin in past 6 months[a] | 9 | 43 | 20.9% | 1 | 40 | 2.5% |
| Other antibiotic use in past 6 months | 30 | 41 | 73.2% | 19 | 34 | 55.9% |
| **Nursing home/elderly home/rehabilitation centre resident** | 16 | 71 | 22.5% | 9 | 69 | 13.0% |

Characteristics are mentioned per category (in bold).
[a]All selective intestinal/oropharyngeal decontamination with colistin.
COLR-EK colistin-resistant E. coli or K. pneumoniae, COLS-EK colistin-susceptible E. coli or K. pneumoniae, IQR interquartile range.

**Table 2 Colistin susceptibility testing policies and methods for Enterobacterales.**

| Policy for colistin susceptibility testing | Colistin susceptibility testing method | | Num. of labs |
|---|---|---|---|
| | Initial test | Confirmation test | |
| Always | VITEK | BMD[a–c] | 7 |
| | | Etest[d] | 1 |
| | | Unknown[c] | 1 |
| Only when considering colistin as treatment | BMD | None | 1 |
| | Unknown | Unknown[c] | 1 |
| In case of a combination of factors | BMD | None | 2 |
| | Etest or BMD | None | 1 |
| | VITEK or BMD | BMD if VITEK was used[a] | 1 |
| | | BMD and when ColR NGS[a] | 1 |
| Only for this study | | | 1 |
| No data available | | | 5 |
| Total | | | 22 |

Confirmation test is performed when: [a]colistin is considered as treatment, [b]in case of a combination of criteria, [c]the isolate is colistin-resistant in the initial test and has certain characteristics or [d]the isolate is colistin-resistant in the initial test.
BMD broth microdilution, ColR colistin-resistant, NGS next-generation sequencing.

**Table 3 Colistin susceptibility testing policies and confirmation of submitted isolates.**

| Used test result for isolate selection | Num. of labs | COLR-EK | | | | COLS-EK | | | |
|---|---|---|---|---|---|---|---|---|---|
| | | Colistin susceptibility testing method[a] | Submitted | Confirmed | % | Colistin susceptibility testing method[a] | Submitted | Confirmed | % |
| Screening test | 15 | Automated[b] | 106 | 61 | 57.5% | Automated[c] | 80 | 61 | 76.3% |
| Confirmation test | 7 | BMD/Etest | 13 | 11 | 84.6% | Automated | 12 | 11 | 91.7% |
| Total | 22 | | 119 | 72 | 60.5% | | 92 | 72 | 78.3% |

Of note: This tabe also includes isolates that are rejected based on other reasons than an incorrect colistin MIC.
[a]Unknown method for the five laboratories with no submitted isolates.
[b]Also two submitted isolates with disk diffusion and two isolates with an Etest.
[c]Also one submitted isolate with disk diffusion.
BMD broth microdilution, COLR-EK colistin-resistant E. coli or K. pneumoniae, COLS-EK colistin-susceptible E. coli or K. pneumoniae.

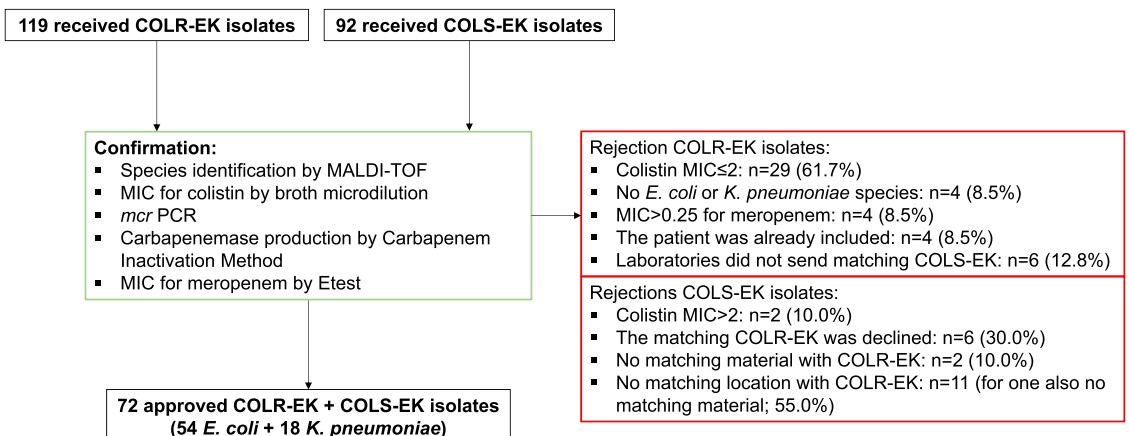

**Fig. 1 Flow chart of received colistin-resistant and colistin-susceptible isolates.** Methods used to decide which isolates met study inclusion criteria are denoted in the green square. Reasons for rejection of isolates for inclusion in the study are denoted in the red squares. COLR-EK: colistin-resistant E. coli or K. pneumoniae, COLS-EK: colistin-susceptible E. coli or K. pneumoniae, MALDI-TOF: matrix-assisted laser desorption/ionisation time-of-flight analyser, MIC: minimum inhibitory concentration, PCR: polymerase chain reaction.

Among 18 participating laboratories that reported numbers of detected COLR-EK, 8243 K. pneumoniae and 45,508 E. coli isolates were identified during the study period of which 81.3% and 82.2% were tested for colistin susceptibility, respectively. Among 6705 tested K. pneumoniae isolates, 42 (0.6%) were found colistin-resistant by participating laboratories. Among 37,392 tested E. coli isolates, 130 (0.3%) were tested colistin-resistant. When combined, 0.4% were tested colistin-resistant from both species.

For selection of COLR-EK and COLS-EK isolates for the current study, some laboratories used other colistin susceptibility testing policies than those used for regular practice. Table 3 shows the used colistin susceptibility testing methods for the submitted isolates and the number of submitted isolates with the percentage of isolates that were confirmed at the RIVM. Fifteen of 22 laboratories based isolate submission only on screening test results (mostly automated tests). As expected, these laboratories had a lower percentage of confirmed isolates compared to laboratories that based submission of isolates on subsequent confirmation tests. In total, 119 presumed COLR-EK and 92 presumed COLS-EK isolates of 17 laboratories were received by the RIVM. Of these, 72 COLR-EK (60.5%) were confirmed to be colistin-resistant and 72 COLS-EK (78.3%) to be susceptible by BMD and were included in the study. This included 18 K. pneumoniae and 54 E. coli per group. Six COLR-EK isolates, including two K. pneumoniae and four E. coli isolates, were only excluded because there was no matching COLS-EK isolate. A flow chart is provided in Fig. 1.

The minimal rate of carbapenem-susceptible colistin-resistant K. pneumoniae and E. coli was 0.01 and 0.03 per 10,000 person years in 21 hospitals with available data, whereas the rate in inpatients in 30 hospitals with available data was 0.20 and 0.41 per 10,000 patient days, respectively.

**Characteristics of isolates**. Thirty-three E. coli and 13 K. pneumoniae COLR-EK/COLS-EK pairs were analysed by NGS. The selection of isolates was based on the best COLR-EK/COLS-EK match and diversity of species and geographical location (more details in 'Methods' section). The genetic composition of the E. coli and K. pneumoniae isolates was highly diverse. For E. coli, there were 41 STs among the 66 isolates, and 23 STs among 26 K. pneumoniae isolates. High-resolution wgMLST corroborated the high degree of genetic diversity (Figs. 2 and 3). It showed the allelic distance between E. coli isolates ranged between 33 and 3524 alleles, with an average of 3123 and a median of 3327 (with 4503 examined genes). The allelic distance between K. pneumoniae isolates ranged between 6 and 3925 alleles, with an average of 3686 and a median of 3782 alleles (with 4978 examined genes).

Only a single genetic cluster of two colistin-resistant K. pneumoniae isolates of patients from different regions submitted by two different laboratories was found.

Microbiological characteristics of all sequenced isolates are shown in Supplementary Data 2 and 3. Among the total of 26 sequenced K. pneumoniae strains, the capsular serotypes

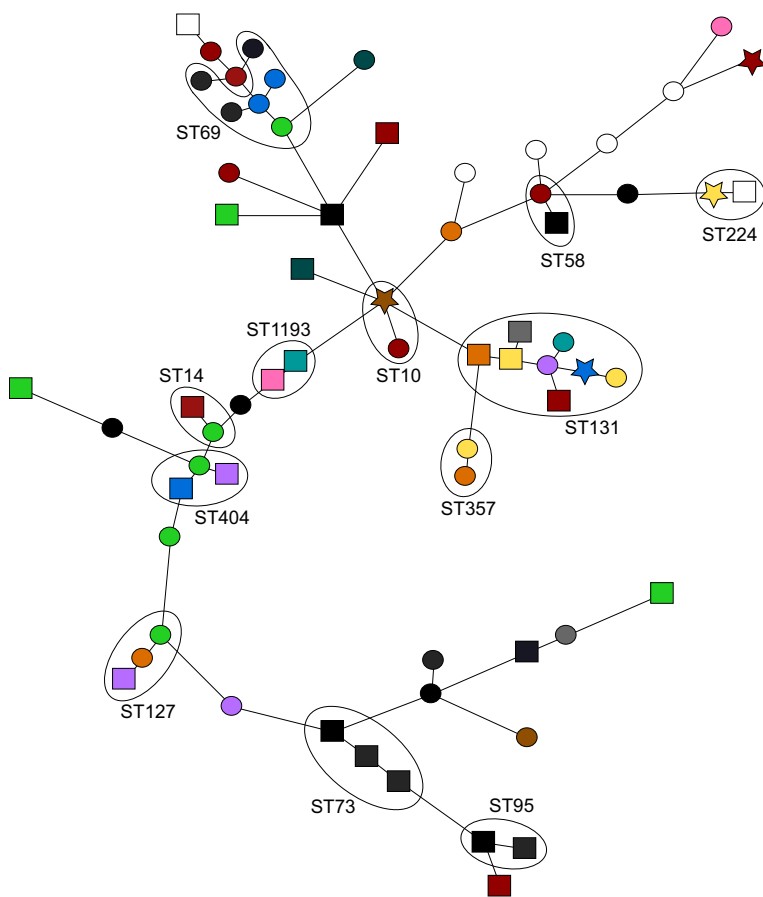

**Fig. 2 Minimum spanning tree of whole genome multilocus sequence typing results of 66 *E. coli* isolates.** Circles/squares/stars represent one isolate, which is connected to the closest relative. The circles represent colistin-susceptible isolates, the squares colistin-resistant isolates and the stars colistin-resistant isolates with *mcr* genes. The length of the lines in between the isolates is proportional to the allelic distance. Groups of isolates with the same classical MLST sequence type are indicated in the figure, e.g. ST131. The colours represent the different participating medical microbiology laboratories. A cluster was defined as ≥2 isolates with an allelic difference of ≤25. There were no clusters observed for *E. coli*. Source data: Supplementary Data files 2 and 3 and the Sequence Read Archive of the National Centre for Biotechnology Information (BioProject ID PRJNA754858).

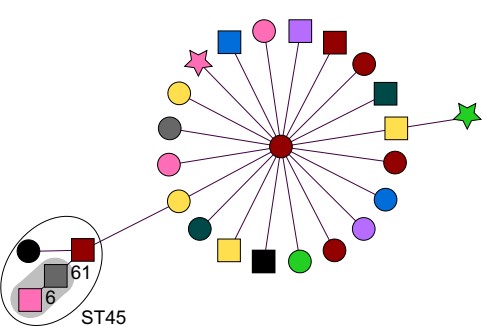

**Fig. 3 Minimum spanning tree of whole genome multilocus sequence typing results of 26 *K. pneumoniae* isolates.** Circles/squares/stars represent one isolate, which is connected to the closest relative. The circles represent colistin-susceptible isolates, the squares colistin-resistant isolates and the stars colistin-resistant isolates with *mcr* genes. The length of the lines in between the isolates is proportional to the allelic distance. For clusters, the allelic differences between the isolates are denoted. Groups of isolates with the same classical MLST sequence type are indicated in the figure (ST45). The colours represent the different participating medical microbiology laboratories. A cluster is indicated by a grey halo and is defined as ≥2 isolates with an allelic difference of ≤20. Source data: Supplementary Data files 2 and 3 and the Sequence Read Archive of the National Centre for Biotechnology Information (BioProject ID PRJNA754858).

(K-antigen) and MLST types were highly diverse. LPS (O-antigen) serotypes were less diverse. The only detected gene encoding a virulence factor for *K. pneumoniae* in this study was yersiniabactin, a siderophore, detected in five COLR-EK isolates (38.5%) and two COLS-EK isolates (15.4%; OR 3.6 (95% CI 0.5–24.2); LR *p* = 0.189). The 66 sequenced *E. coli* strains were also highly diverse in serotypes and MLST types. The well-known multi-drug resistant ST131 was observed in eight isolates (*n* = 5 COLR-EK of which one with *mcr-1*, *n* = 3 COLS-EK), of which seven were derived from urine samples. Numerous different genes encoding virulence factors were observed in *E. coli* isolates, but there were no significant differences between the colistin-resistant and -susceptible group. When only urine samples were included (*n* = 46; 23 per group), there were also no significant differences.

**Colistin resistance**. The median colistin MIC by BMD was 8.0 mg/L (IQR 4.0–12.0) in the COLR-EK and 0.25 mg/L (IQR 0.25-0.5) in the COLS-EK group.

*Colistin resistance caused by* mcr *genes*. Of 72 COLR-EK isolates, *mcr* genes were found in seven (9.7%) isolates. Of 18 colistin-resistant *K. pneumoniae* isolates, two contained *mcr-9* and one *mcr-1*, whereas, of 54 colistin-resistant *E. coli* isolates, four contained *mcr-1*. The isolates containing *mcr-1* had a colistin MIC of 2, 4 or 16 mg/L with BMD, whilst both *mcr-9* isolates had a

colistin MIC of 32 mg/L. We attempted to cure the latter two isolates from their *mcr-9* genes by repeated subculture and were successful for one. Despite the loss of the *mcr-9* gene, the isolate retained its high MIC for colistin. No new chromosomal mutations in the *pmrAB*, *phoPQ*, *crrB* and *mgrB* genes were found after curing. NGS and TGS of the isolate revealed that 39,237 base pairs including the *mcr-9* gene were deleted from the plasmid. Molecular characterisation of the five *mcr-1* positive isolates, revealed that *mcr-1* of one isolate (MIC = 4 mg/L) was two base pairs shorter than the reference (the first amino acid started with threonine instead of methionine). Whether the gene was still functional is unknown.

Supplementary Data 4 shows the results of NGS and hybrid assembly of the isolates with *mcr* genes. The isolates with *mcr* genes all had different MLST types, capsular serotypes and LPS serotypes, which indicates different genetic backgrounds. Figure 4 shows characteristics of the five *mcr*-containing plasmids with available hybrid assembly data (IDs starting with RIVM) and previously found plasmids from the NCBI database that highly resembled them (63–91% identity and 84–100% query). The plasmids with *mcr* genes from this study contained varying % G + C content and varying replicon families. More specifically *mcr-1* plasmids included replicons IncHI2 or no replicon, and the *mcr-9* plasmids replicons IncHI2, IncHI2A and IncR. The *mcr* plasmids were unrelated (<64% identity). The *mcr-9* plasmids from two *K. pneumoniae* isolates appeared different, since the AMR genes and the plasmid size differed substantially, though they both contained ESBL gene *bla*CTX-M-9. The *mcr-1* plasmids of *E. coli* did not contain other AMR genes and differed in size, whereas the *mcr-1* plasmid of *K. pneumoniae* contained several other AMR genes and was considerably larger than the *mcr-1* plasmids from *E. coli*. The previously identified plasmids from the NCBI database were found in Spain, USA, Qatar and Brazil. Two patients with isolates harbouring both *mcr-1* visited a foreign country in the previous 6 months (Egypt and an unknown country) and they had both been hospitalised in that country.

*Chromosomal mutations potentially associated with colistin resistance.* Table 4 shows the isolates with *mcr* genes and/or chromosomal mutations in the *pmrAB*, *phoPQ*, *mgrB* and *crrB* genes, that are potentially associated with colistin resistance. Chromosomal mutations that were present in COLS-EK isolates only or in both COLS-EK and COLR-EK isolates were removed and the entire list of chromosomal mutations can be found in Supplementary Data 5. Of 46 sequenced COLR-EK isolates, seven

had an *mcr* gene (15.2%) and 19 (41.3%) had a chromosomal mutation potentially associated with colistin resistance. Of the latter, two also had an *mcr* gene, i.e. one with *mcr-1* and one with *mcr-9*. Among the 13 sequenced colistin-resistant *K. pneumoniae* isolates, eight isolates with chromosomal mutations were found. Mutations were mainly found in the *mgrB* gene, with the majority being insertions. Among the 33 sequenced colistin-resistant *E. coli* isolates, 11 chromosomal mutations were found, mainly in the *pmrB* gene. In total, 3 of 13 (23.1%) colistin-resistant *K. pneumoniae* and 19 of 33 (57.6%) colistin-resistant *E. coli* isolates had unexplained resistance to colistin.

**Antimicrobial resistance for other antibiotics**. There were significantly more MDRO in the COLR-EK group (17/72 (23.6%)) compared to the COLS-EK group (6/68 (8.8%)), with an OR of 3.9 (95% CI 1.3–12.2; LR *p* = 0.019). Of COLR-EK isolates, 19.7% (14/71) was ESBL-producing, which was significantly more than the 7.4% (5/68) of the COLS-EK group (OR 3.5; 95% CI 1.1–11.6; LR *p* = 0.037). For *E. coli*, the 33 sequenced COLR-EK isolates contained a median of 3 (IQR 1:7) AMR genes per isolate and 1 (IQR 1:4) per isolate in the COLS-EK group. Similarly, the 13 *K. pneumoniae* isolates had a median of 5 (IQR 4:11) AMR genes per isolate for the COLR-EK isolates and a median of 4 (IQR 4:7) for the COLS-EK isolates.

## Discussion

This study showed that colistin resistance among Enterobacterales cultured from patients is present but uncommon in the Netherlands. Patients with COLR-EK isolates had more frequently used colistin in SDD/SOD, compared to COLS-EK patients. WgMLST revealed there was no comprehensive dissemination of highly similar colistin-resistant strains. Only seven (9.7%) of the COLR-EK isolates carried *mcr* genes, including five with *mcr-1* and two with *mcr-9*, whereas 19 (41.3%) sequenced COLR-EK isolates had a chromosomal mutation potentially associated with colistin resistance. Interestingly, the *mcr-9* gene did not elicit phenotypical colistin resistance in our cohort. Furthermore, the COLR-EK group contained more MDRO compared to the COLS-EK group. This could potentially be explained by more previous antibiotic use (at any moment in time) in the COLR-EK group and co-localisation of AMR genes on *mcr* plasmids.

In this study, 0.4% of tested isolates were found colistin-resistant by participating laboratories. This was in complete accordance with ISIS-AR data. In the same study period, 40 laboratories that

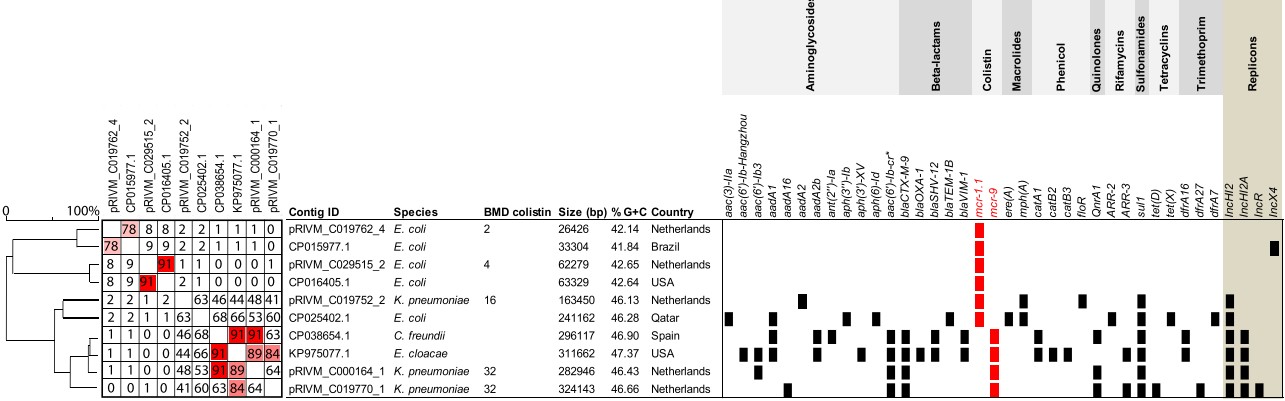

**Fig. 4 Comparison of plasmids containing *mcr* genes.** Contig IDs that start with 'pRIVM' are from plasmids of isolates collected in the current study. The other isolates are the closest resembling plasmids from the NCBI database. Percentages identity are depicted in the square on the left. Source data: Supplementary Data file 4 and GenBank of the National Centre for Biotechnology Information (BioProject ID PRJNA754858). BMD: broth microdilution, bp: base pairs, USA: United States of America, % G + C: guanine-cytosine content.

**Table 4 Overview of isolates with *mcr* genes or non-silent chromosomal mutations, potentially involved in colistin resistance.**

| Isolate ID | *mcr* | *pmrA* | *pmrB* | *mgrB* | *crrB* | *phoP* | *phoQ* |
|---|---|---|---|---|---|---|---|
| *K. pneumoniae* | | | | | | | |
| RIVM_C019776 | No | | | First 39 bp are absent | | | |
| RIVM_C019778 | No | | | P74: insertion ISEc68 | | | |
| RIVM_C019785 | No | | | G37S(G109A) | | | |
| RIVM_C019837 | No | | T157P(A469C) | | | | |
| RIVM_C000156 | No | | | Absent | 955 bp deletion | | |
| RIVM_C000119 | No | | | P46: insertion IS-like el | | | |
| RIVM_C019878 | No | | | P70: insertion IS-like el | | | |
| RIVM_C019770 | *mcr-9* | | | C39Y(G116A) | | | |
| RIVM_C019752 | *mcr-1.1* | | | | | | |
| RIVM_C000164 | *mcr-9* | | | | | | |
| *E. coli* | | | | | | | |
| RIVM_C019737 | No | | V128E(T382A) | | | | |
| RIVM_C019749 | No | | T159M(C475T) | | | | |
| RIVM_C019767 | No | | V91E(T272A) | | | | |
| RIVM_C019769 | No | | | | | | E464D(G1392T) |
| RIVM_C019789 | No | | P22: extra GCG(aa A) | | | | |
| RIVM_C019808 | No | | L105Q(T314A) | | | | |
| RIVM_C019825 | No | N67K(C200A + C201A); D68E(C204A) | M4I(G12C) | | | | |
| RIVM_C019864 | No | | V91E(T272A + A273G) | | | | |
| RIVM_C019866 | No | | First 48 bp are absent | | | V88A(T263C) | |
| RIVM_C028932 | No | | M4I(G12C) | | | | |
| RIVM_C029515 | *mcr-1.1* | | | | | I175F(A523T) | |
| RIVM_C000121 | *mcr-1.1* | | | | | | |
| RIVM_C019762 | *mcr-1.1* | | | | | | |
| RIVM_C019792 | *mcr-1.1* | | | | | | |

Point mutations are indicated by the amino acid substitution (nucleotide substitution) with the number representing the location. Silent mutations and mutations that were present in (both colistin-resistant and) colistin-susceptible isolates were not included in this table.
*aa* amino acid, *bp* base pairs, *el* element, *MIC* minimum inhibitory concentration, *p* position.

provided results from screening tests and/or confirmation tests for ISIS-AR identified 0.4% (290/71,839) of tested *K. pneumoniae* and *E. coli* isolates as colistin-resistant. This indicates that the laboratories that participated in this study were a good representation of all laboratories in the Netherlands.

This study provides important insights into the epidemiology of colistin resistance in the Netherlands. Most previous studies examining colistin resistance in humans in the Netherlands focused on the *mcr-1* gene[43–45], specific patient populations[46–51] or travellers[52–54]. Similar to the current study, a higher percentage of colistin resistance caused by chromosomal mutations compared to *mcr* genes was also found by Bourrel et al.[55], who screened patients in six Parisian hospitals for rectal carriage of colistin-resistant *E. coli*.

Interestingly, we found the presence of *mcr-9* not to be associated with phenotypical resistance. Some other studies also reported a minor impact of *mcr-9* on the colistin MIC[56–58], although the MIC may be increased in the presence of colistin by the higher expression of qseC and qseB genes, encoding a histidine kinase sensor and its cognate partner of a two-component regulatory system that regulates *mcr-9* expression[59]. The isolate in our study, that was cured from the *mcr-9* gene, also had a mutation in the *mgrB* gene, which led to an amino acid difference (C39Y). Possibly, this mutation caused colistin resistance, but information on this mutation is scarce[18,60].

We found several chromosomal mutations presumed to be associated with colistin resistance. For *K. pneumoniae*, most chromosomal mutations that were only found in colistin-resistant isolates were verified in experimental settings in previous studies. The most frequent observed chromosomal mutation was an insertion in the chromosomal *mgrB* gene, which is often described as cause of colistin resistance in *K. pneumoniae*[60–65]. For *E. coli*, the involvement of chromosomal mutations in colistin resistance is mostly not confirmed by laboratory experiments[18].

We found that previous colistin use was associated with colistin resistance, which is in accordance with several previous studies[11]. In our study, colistin was used only as component of SDD/SOD to prevent development of infections in specific patient groups, such as patients with a haematological disease or patients staying at the intensive care. Though most studies showed SDD/SOD is effective in reducing the number of infections in specific patients groups[66], it

is only used in a small number of countries[67,68]. The use of SDD/SOD remains a controversial topic mainly because of the potential selection and spread of multi-drug resistant bacteria. Numerous studies found no difference in presence of colistin resistance during SDD/SOD compared to controls without SDD/SOD[48,69–73], but there are also studies that suggest an increase in colistin resistance during SDD/SOD[50,74]. Most studies examined colistin resistance during or shortly after the use of SDD/SOD, but our data suggest that a longer period is needed to detect colistin-resistant bacteria, which may be relevant for future clinical trials.

Another factor found to be associated with colistin resistance in the matching-adjusted analysis in this study was malignancy, but this effect disappeared after correction for colistin use. Colistin is also used as SDD in patients with haematological malignancies, which may explain the increased colistin resistance in these patients. A previous study suggested that the use of colistin may be involved in the pathogenesis of colorectal cancer by stimulating the production of colibactin in procarcinogenic bacteria[75], but the design of our study does not provide sufficient data to evaluate this. We examined the presence of colibactin-producing genes in our 66 sequenced *E. coli* isolates and found a *pks*-island in 21 isolates, of which 12 were colistin-resistant. However, none of the isolates contained a complete *pks*-island.

The detection of 72 confirmed colistin-resistant isolates from 17 of 22 participating laboratories in this study, with an underestimation of the occurrence due to the exclusion of CRE and restricted testing in participating laboratories, shows that colistin-resistant isolates are present in the Netherlands and therefore there is a need for surveillance of routine diagnostic AST results of colistin. This is further emphasised by the increasing colistin resistance worldwide and has been recognised by ECDC[8–10]. As a result, standardisation of colistin susceptibility testing will become more important. Unfortunately, colistin-susceptibility testing with EUCAST and CLSI-recommended BMD[31] is frequently not part of the routine AST panel for Enterobacterales[3], as also observed in this study. Furthermore, only 52.9% of the participating laboratories screened all Enterobacterales for colistin resistance. These laboratories used automated systems as screening test, which are inferior compared to BMD[15]. Unfortunately, many laboratories are reluctant to use BMD for all *E. coli* and *K. pneumoniae* isolates because the application is very labour-intensive. However, high very major

error rates (producing false susceptible results) are reported for automated systems and they are therefore not suitable as screening test. Major error rates (producing false-resistant results) are low[15,16]. In contrast, 24.4% (29/119) of by participating laboratories presumed COLR-EK isolates were rejected in this study due to colistin MIC ≤ 2 mg/L and only 2.2% (2/92) of COLS-EK isolates were rejected due to MIC > 2 mg/L. The majority of the laboratories sent in isolates for this study based on automated testing methods. This suggests a high major error rate of automated systems in Dutch MMLs. In addition to surveillance of routine diagnostic AST, testing of all *E. coli* and *K. pneumoniae* isolates for colistin resistance with methods that are more reliable than automated testing methods, is needed. Possibly, the Rapid Polymyxin NP test may be a good alternative for automated testing, since it is less labour-intensive. It detects glucose metabolisation associated with bacterial growth when colistin or polymyxin B is present and can be completed in <2 h[60,76].

This study provided important insights into the presence of COLR-EK in humans in the Netherlands, the current susceptibility testing policies, distribution of strains and genomic characteristics of colistin-resistant Enterobacterales. However, there were also a few limitations. This study may have underestimated the true incidence of colistin resistance, since carbapenem-resistant Enterobacterales (CRE) isolates were not included. However, data from ISIS-AR showed that only 1.4% of colistin-resistant isolates were carbapenem-resistant with an MIC > 8 mg/L. Second, several laboratories only tested a selection of isolates for colistin resistance and provided a maximum of five COLR-EK isolates per laboratory. Finally, persons that filled in the questionnaires were not blinded for colistin susceptibility testing results and therefore data on colistin use that were missing not at random cannot be ruled out.

In conclusion, COLR-EK isolates are present in the Netherlands and colistin resistance is caused by *mcr* genes in a minority of isolates. This study suggests that diverse COLR-EK populations for both *E. coli* and *K. pneumoniae* are present. In the future, surveillance of routine diagnostic AST with detection of *mcr* genes and molecular typing should be implemented to monitor and control the occurrence and spread of colistin resistance and *mcr* genes. For this, testing of all *E. coli* and *K. pneumoniae* isolates with more reliable testing methods by local laboratories is needed.

## Data availability

Raw NGS sequence data of all sequenced isolates were deposited in the Sequence Read Archive and plasmids with *mcr* genes in GenBank of the National Centre for Biotechnology Information (NCBI) under BioProject ID PRJNA754858. Microbiological characteristics of all sequenced isolates are shown in Supplementary Data 2 and 3. The epidemiological dataset analysed during the current study is uploaded in the repository Open Science Framework (https://osf.io/vj6pb/?view_only=301241c0502e4b53ac12050712b36bfe). Laboratory and hospital data from questionnaires included in this study and from ISIS-AR are only available from the corresponding author upon reasonable request and with permission of the participating laboratories because it concerns sensitive information concerning the production, workflow and policies of medical microbiological laboratories and hospitals.

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

## Acknowledgements
We thank all the participating laboratories and the members of the ColRE survey study group. Furthermore, we thank Marga van Santen-Verheuvel and Han van der Heide for providing support in the analysis of chromosomal mutations, performing laboratory work and creating the database. We thank Sam Nooij for the colibactin-analysis. We thank ECDC for initiating the CCRE survey, which provided the framework for this study. Finally, we thank Rony Zoetigheid and Wieke Altorf-van der Kuil for providing data from the Dutch registration ISIS Antimicrobial Resistance of the RIVM Centre for Infectious Disease Control. ISIS-AR is maintained by the RIVM, the Dutch medical microbiology laboratories and the professional association of medical microbiologists NVMM.

## Author contributions
The study was conceived and supervised by L.M.S. and E.J.K. K.E.W.V. coordinated the study, performed part of the analysis and drafted the study protocol and the manuscript. A.D.H. coordinated the laboratory experiments. F.L. and A.D.H. performed laboratory experiments, TGS and hybrid assemblies. S.W. analysed the NGS data. A.P.A.H. analysed *mcr* plasmid data. A.P.A.H., D.W.N., P.B., A.F.S., S.C.D.G. and C.C.H.W. aided in composing the study protocol. J.J.G. provided advise on the statistics. C.C.H.W., A.F.S. and S.C.D.G. provided advise on epidemiological analysis. All authors critically reviewed the manuscript. The ColRE survey study group collected and sent the isolates and provided isolate, patient, hospital and laboratory data.

## Competing interests
The authors declare no competing interests.

## Additional information

## ColRE survey consortium

Karen Heemstra[5], Saara Vainio[6], Alewijn Ott[7], Steve de Jager[8], Fleur Koene[9], Vishal Hira[10], Nathalie van Burgel[11], Anouk Muller[12], Karolien Nagtegaal-Baerveldt[13], Coby van der Meer[14], Rik van den Biggelaar[15], Oscar Pontesilli[16], Suzan van Mens[17], Wouter van den Bijllaardt[18], Eva Kolwijck[19], Ron Bosboom[20], Ine Frénay[21], Annemarie van 't Veen[22], Annet Troelstra[23], Greetje Kampinga[24] & Karin van Dijk[25]

[5]Alrijne zorggroep, Leiden, The Netherlands. [6]St. Antonius Hospital, Nieuwegein, The Netherlands. [7]Certe, Groningen, The Netherlands. [8]Comicro, Hoorn, The Netherlands. [9]Public Health Laboratory, Amsterdam, The Netherlands. [10]Groene Hart Hospital, Gouda, The Netherlands. [11]Haga Hospital, Den Haag, The Netherlands. [12]Haaglanden Medical Centre, Den Haag, The Netherlands. [13]Ikazia Hospital, Rotterdam, The Netherlands. [14]Izore, Leeuwarden, The Netherlands. [15]Laboratory for Medical Microbiology and Immunology, Tilburg, The Netherlands. [16]Maasstad Laboratory, Rotterdam, The Netherlands. [17]Maastricht University Medical Centre+, Maastricht, The Netherlands. [18]Microvida, Breda, The Netherlands. [19]Radboud University Medical Centre, Nijmegen, The Netherlands. [20]Medical Microbiology and Immunology Laboratory, Arnhem, The Netherlands. [21]Regional Laboratory Medical Microbiology, Dordrecht, The Netherlands. [22]Saltro, Utrecht, The Netherlands. [23]University Medical Centre Utrecht, Utrecht, The Netherlands. [24]University Medical Centre Groningen, Groningen, The Netherlands. [25]Amsterdam University Medical Centres, Location VUmc, Amsterdam, The Netherlands.

