## [Peer Review File · Communications Medicine]

Reviewers' comments:

Reviewer #1 (Remarks to the Author):

This is a very nice and technically well done study.

Results are interesting and address the molecular epidemiology in a satisfactory way.

some data are very close to other data that have been previously reported

Reviewer #2 (Remarks to the Author):

In their manuscript, Vendrik et al provide an overview of colistin resistant/susceptible isolates collected across the Netherlands over a ~9 month period. This was an interesting study and most of my comments mainly relate to providing more specific details or numbers as outlined below. I do however wanted to query if there was any reason why the authors didn't generate phylogenetic trees for both E. coli and Kp, providing higher resolution between the relationship of strains? It would be insightful to highlight how many STs were detected and whether any STs clusters encompassed both colistin resistant and susceptible isolates. This information along with the referring lab and mechanism of resistance can all be annotated next to the phylogenies to provide a comprehensive overview of if and how colistin-resistant/susceptible isolates are related, and highlight that the mcr genes are circulating between non-related strains. Further, are the authors able to make some commentary on whether there are STs that were common in their colistin-resistant isolates and whether these same STs are also being reported with colistin resistance in neighbouring European countries?

Additional comments:

Lines 45-47: can the authors be more specific with percentage ranges? i.e. >10% is representative of a wide range, and would also be good to provide some indication of prevalence for E. coli.

Line 65: statement is vague; can the authors elaborate on what these methods are for testing and how the results are unreliable?

Line 68: at the very least, authors should specify what each of these genes encode and maybe also how these mutations give rise to resistance. Are the authors also able to clarify how frequently these mutations are detected in Kp (and E. coli?)

Line 74: how abundant?

Lines 79-80: please describe briefly this outbreak for some context - i.e. how many isolates? Hospitals? Time span or year of study?

Line 92: given that the results section comes before the methods, it would be good to briefly describe the inclusion criteria here.

Line 95: what is RIVM? Abbreviation needs to be defined

Lines 96-97: The authors mention here that some laboratories sent more than the maximum of n=5 colistin R isolates and these were included when the inclusion criteria were met; was there a reason why a limit of n=5 was set and does discarding this limit for some regions create some issue/bias with the dataset?

Lines 145-146: briefly state how strains were selected for sequencing to provide context and refer to

methods for more details

Lines 147-149: please specify how many alleles are included in the wgMLST schemes in order to understand how diverse the mean allelic differences are.

Line 177: where does the 2bp deletion occur? i.e. how many bp into the gene? Or specify whereabouts the mutation occurs i.e. towards end of gene etc.

Line 190: how similar (i.e. % identity/coverage) were these plasmids to those detected in these other countries?

Lines 195-204: Can the authors specify how many isolates of Kp/E. coli had unexplained resistance, if any? i.e. no mcr gene or chromosomal mutations detected

Lines 210-213: are any of the genes reported for the resistant/susceptible groups core AMR genes? This should be clarified

Line 216: the authors mention that their study shows that colistin resistance is not uncommon.. perhaps reword this; correct me if I'm wrong but I feel as though a detection rate of 0.4% is not particularly common.

Lines 222-224: partially also due to co-localisation of AMR genes on the same plasmid backbones mobilising mcr?

Line 239: what functions do the qse genes serve and please also clarify how this relates to colistin uptake/resistance?

Line 267: unclear what is meant by "all pks-islands were not completely present" - isn't there only a single pks island? Or do the authors mean that some of the island/operon is missing/partially deleted?

Line 288: suggest briefly describing what the rapid polymyxin NP test entails and whether it is already used in standard lab AST practices?

Lines 354-355: unclear by what is meant by "good representativeness of both E. coli and Kp"?

Line 356: Can the authors specify here what sequencing platforms/library preparations they used for NGS and TGS? They specify ONT sequencing in line 370 but don't clarify the platform used to generate their short reads.

Lines 356-369: perhaps re-order this paragraph to start with a description of NGS sequencing and library preparation before going into methodology involved in using the sequencing data

Line 364: citations and/or web links missing for VirulenceFinder, SerotypeFinder and Kleborate

Lines 375-377: can the authors clarify why they focused on the mcr-9 isolates and not say, the mcr-1?

Lines 379-284: Can the authors clarify why they used read mapping to ascertain mutations when they could have directly interrogated their assemblies for these gene sequences and then align? Does this yield any differences to what was observed using the read mapping approach?

Figure 1: suggest adding in a breakdown of the 72 matched isolates by species i.e. n=X E.coli vs. Kp

Figure 3: what do the '6' and '61' in the figure denote?

Figure 4: suggest having the AMR genes and AMR class labels slanted on a 45degree angle or so; currently difficult to read

Minor comments:

Line 61: grammatical error: "had tested colistin-resistant"

Line 169: typo "pneumoniae"

Line 287: typo "than"

Dear reviewers,

We would like to thank the reviewers for the time and efforts that were taken to review our manuscript. Many of the comments were useful and we feel that the manuscript has improved considerably after addressing the suggestions of the reviewer.

Please find below our reply to the individual comments. Changes are made in the revised manuscript and, if not, reasons for rebuttal are given. Due to the added information as requested by the reviewers, the maximum permitted number of words was mildly exceeded with 170 words.

We have highlighted the changes in the manuscript by track changes. If we refer to lines in the manuscript in our replies, we refer to the manuscript with track changes.

Yours sincerely,

On behalf of all co-authors of this manuscript,

Mrs. Karuna Vendrik MD
PhD-student
National Institute for Public Health and the Environment
The Netherlands
Karuna.vendrik@rivm.nl
0031-620530040

Referee expertise:

Referee #1: Genomic epi, AMR, colistin resistance, clinical

Referee #2: Microbiology/microbial genomics, AMR

Reviewers' comments:

Reviewer #1 (Remarks to the Author):

This is a very nice and technically well done study.

Results are interesting and address the molecular epidemiology in a satisfactory way.

some data are very close to other data that have been previously reported

Reviewer #2 (Remarks to the Author):

In their manuscript, Vendrik et al provide an overview of colistin resistant/susceptible isolates collected across the Netherlands over a ~9 month period. This was an interesting study and most of my comments mainly relate to providing more specific details or numbers as outlined below. I do however wanted to query if there was any reason why the authors didn't generate phylogenetic trees for both E. coli and Kp, providing higher resolution between the relationship of strains? It would be insightful to highlight how many STs were detected and whether any STs clusters encompassed both colistin resistant and susceptible isolates. This information along with the referring lab and mechanism of resistance can all be annotated next to the phylogenies to provide a comprehensive

overview of if and how colistin-resistant/susceptible isolates are related, and highlight that the *mcr* genes are circulating between non-related strains.

Reply: If we understand correctly, the reviewer wonders why we did not provide phylogenetic trees based on MLST to provide a higher resolution relationship between the strains. However, we provided high resolution phylogenetic trees based on wgMLST. In the classical MLST only seven genes are used, whereas wgMLST uses 4,503 (*E. coli*) and 4,978 (*K. pneumoniae*) genes. As wgMLST provides a much higher resolution, we utilised this approach. Because only a small selection of isolates is included in this study, no statements on evolution could be included. Detailed information on the isolates is provided in the supplementary table, including MLST STs. The diversity within the STs was high. There was only one cluster including two colistin-resistant strains with ST45. To include results obtained with classical MLST, we changed the text in the Results section accordingly in lines 206-213 and added information on the MLST STs and *mcr* genes in figures 2 and 3.

Further, are the authors able to make some commentary on whether there are STs that were common in their colistin-resistant isolates and whether these same STs are also being reported with colistin resistance in neighbouring European countries?

Reply: As also mentioned in the manuscript, diversity within the STs was high. For colistin-resistant *K. pneumoniae* the most common ST was ST45, but this ST was found in only in three of the 13 colistin-resistant *K. pneumoniae* isolates. None of these carried an *mcr* gene. For colistin-resistant *E. coli*, the most common ST was ST131, which was reported in only five out of the 33 colistin-resistant *E. coli* isolates. One of these had an *mcr* gene. However, there were also three ST131 isolates among the colistin-susceptible isolates. The wgMLST-based genetic diversity within the ST131 group was high, with a pairwise allelic distance ranging from 113 to 587 alleles. We decided not to discuss STs that are more common in colistin-resistant isolates and whether these are found in other European countries in the manuscript, because the STs are very diverse and even the most common ST that are mentioned above are observed in less than a quarter of the isolates. Furthermore, as far as we know, no systematic studies on the prevalence and genomic characteristics of colistin-resistant *E. coli* and *K. pneumoniae* isolates in a similar human clinical setting have been performed before. Our study was part of a pan-European multicentre study on colistin- and carbapenem-resistant Enterobacteriales (see lines 385-387 and <https://www.ecdc.europa.eu/en/publications-data/ecdc-study-protocol-genomic-based-surveillance-carbapenem-resistant-andor>), but we have not seen any data reported on STs in Europe yet that can be included in our manuscript.

Additional comments:

Lines 45-47: can the authors be more specific with percentage ranges? i.e. >10% is representative of a wide range, and would also be good to provide some indication of prevalence for *E. coli*.

Reply: We have added specific percentages to the new version of the manuscript in lines 81-85: *'The mean resistance percentage for carbapenem among Klebsiella pneumoniae isolates in Europe is 7.9%, with some countries reporting resistance percentages between 25 to 50% or ≥50%. It is observed in only 0.3% of Escherichia coli isolates.'*

Line 65: statement is vague; can the authors elaborate on what these methods are for testing and how the results are unreliable?

Reply: To clarify this, we added this sentence to the manuscript in lines 104-109: *'Broth microdilution is the gold standard method, but is labour-intensive and time-consuming. Methods such as disk diffusion and agar dilution produce unreliable results due to the large molecular size of colistin making it poorly diffusible through agars. Furthermore, many laboratories use*

automated antimicrobial susceptibility testing (AST) systems with high very major error rates (producing false susceptible results).'

Line 68: at the very least, authors should specify what each of these genes encode and maybe also how these mutations give rise to resistance. Are the authors also able to clarify how frequently these mutations are detected in Kp (and E. coli?)

We understand the reviewers point to provide more information on the function of the genes and the contribution of mutations to colistin resistance. To provide a general explanation on the function of the genes, we have made changes to the manuscript in lines 110-115: ‘For K. pneumoniae, mutations in the chromosomally located *pmrA/prmB*, *phoPQ*, *mgrB* and *crrB* genes have been intensively studied. Mutations in these genes lead to the upregulation of the modification of lipid A in lipopolysaccharide (LPS). This modification leads to decreased negative charge of the bacterial membrane impairing the interaction between colistin and the LPS.’

However, in our opinion it is outside of the scope of this manuscript to discuss the function of the genes in detail and to assess via an extensive literature review how frequently mutations are detected in *K. pneumoniae* and *E. coli*. The function of the genes are described in the article of Gogy et al (doi: 10.3389/fmed.2021.677720), to which we refer in the manuscript.

Line 74: how abundant?

Reply: To clarify this, we added this sentence to the manuscript in lines 122-123: ‘A study that examined 457 *mcr-1*-positive *Enterobacterales* isolates from 31 different countries, found 411 *E. coli* isolates (89.9%).’

Lines 79-80: please describe briefly this outbreak for some context - i.e. how many isolates? Hospitals? Time span or year of study?

Reply: To provide more details of the outbreak, we have added some words to the sentence in lines 129-131: ‘One outbreak with six patients from a hospital and nursing home with a colistin-resistant carbapenemase-producing *K. pneumoniae* in 2013 has been described.’

Line 92: given that the results section comes before the methods, it would be good to briefly describe the inclusion criteria here.

Reply: We added the inclusion criteria for COLR-EK and COLS-EK isolates in lines 146-153: ‘inclusion criteria for this study: *ColR-EK* with a colistin MIC > 2 mg/L and/or a *mcr*-gene, a meropenem MIC ≤ 0.25 mg/L and no carbapenemase production. *COLR-EK* isolates obtained from 72 patients were confirmed colistin-resistant at the Dutch National Institute for Public Health and the Environment (RIVM) and were included in this study. Colistin-susceptible *E. coli* or *K. pneumoniae* (*COLS-EK*) isolates from 72 control patients were included, with a colistin MIC ≤ 2 mg/L, no *mcr*-gene and a meropenem MIC ≤ 0.25 mg/L, and matched for patient location, material of origin and bacterial species.’

Line 95: what is RIVM? Abbreviation needs to be defined

Reply: We have added the definition of the RIVM in lines 149-150.

Lines 96-97: The authors mention here that some laboratories sent more than the maximum of n=5 colistin R isolates and these were included when the inclusion criteria were met; was there a reason why a limit of n=5 was set and does discarding this limit for some regions create some issue/bias with the dataset?

Reply: The reason for the maximum of five were the otherwise high costs and time-burden for our scientists, physicians and technicians. Some laboratories did not count their sent isolates

and therefore exceeded the maximum number of isolates. To compensate for their efforts, we decided to include these isolates. This means these isolates were included at random and therefore we do not assume this has introduced any bias.

Lines 145-146: briefly state how strains were selected for sequencing to provide context and refer to methods for more details

Reply: We included this sentence in the manuscript in lines 205-206: ‘The selection of isolates was based on the best COLR-EK/COLS-EK match and diversity of species and geographical location (more details in methods section).’

Lines 147-149: please specify how many alleles are included in the wgMLST schemes in order to understand how diverse the mean allelic differences are.

Reply: We added ‘(with 4503 examined genes)’ and ‘(with 4978 examined genes)’ in lines 211 and 213.

Line 177: where does the 2bp deletion occur? i.e. how many bp into the gene? Or specify whereabouts the mutation occurs i.e. towards end of gene etc.

Reply: To provide more information on the 2bp deletion, the following was added to the manuscript in line 246-247: ‘(the first amino acid started with threonine instead of methionine)’.

Line 190: how similar (i.e. % identity/coverage) were these plasmids to those detected in these other countries?

Reply: The % identity is depicted in the square on the left in figure 4. To clarify this, we added the X-axis above the square in figure 4 and added this to the legend of figure 4: ‘Percentages identity are depicted in the square on the left.’ Furthermore, we added these words to the manuscript in line 252: ‘(63-91% identity and 84-100% query)’

Lines 195-204: Can the authors specify how many isolates of Kp/E. coli had unexplained resistance, if any? i.e. no mcr gene or chromosomal mutations detected

Reply: This sentence was added to the manuscript in lines 275-277: ‘In total, 3 of 13 (23.1%) colistin-resistant K. pneumoniae and 19 of 33 (57.6%) colistin-resistant E. coli isolates had unexplained resistance to colistin.’

Lines 210-213: are any of the genes reported for the resistant/susceptible groups core AMR genes? This should be clarified

Reply: For the isolates that have mcr genes, this can be found in table 4. In table 4 the study IDs that start with a ‘c’ indicate chromosome contigs and the study IDs that start with a ‘p’ indicate plasmid contigs. This is clarified in more detail in the legend of the table. For the other isolates, this is less relevant in our opinion and therefore this is not described in the manuscript.

Line 216: the authors mention that their study shows that colistin resistance is not uncommon.. perhaps reword this; correct me if I’m wrong but I feel as though a detection rate of 0.4% is not particularly common.

Reply: We agree that the wording is not correct. It was more than we had expected in the Netherlands. We have changes ‘not uncommon’ into ‘present but uncommon’ in lines 289-290 of the manuscript and in lines 63 and 346 we have changed ‘not rare’ into ‘present but uncommon’ or ‘present’.

Lines 222-224: partially also due to co-localisation of AMR genes on the same plasmid backbones mobilising *mcr*?

Reply: We agree that this may also play a role, although there were only seven isolates with plasmids containing *mcr* genes, of which two did not contain any other AMR-genes. We have added ‘and co-localisation of AMR genes on *mcr* plasmids’ to the manuscript in line 298.

Line 239: what functions do the *qse* genes serve and please also clarify how this relates to colistin uptake/resistance?

Reply: We have added this to the manuscript in lines 313-315: ‘, encoding a histidine kinase sensor and its cognate partner of a two-component regulatory system that regulates *mcr-9* expression.’

Line 267: unclear what is meant by “all *pks*-islands were not completely present” - isn’t there only a single *pks* island? Or do the authors mean that some of the island/operon is missing/partially deleted?

Reply: We agree this needs to be clarified. We have changed the sentence in lines 342-343 of the manuscript into: ‘However, none of the isolates contained a complete *pks*-island.’

Line 288: suggest briefly describing what the rapid polymyxin NP test entails and whether it is already used in standard lab AST practices?

Reply: We added a short description of the test in lines 365-366: ‘It detects glucose metabolism associated with bacterial growth when colistin or polymyxin B is present and can be completed in < 2 hours.’ However, in our opinion, this is beyond the scope of this article and therefore we chose not to elaborate on this further. It is a relatively new method and we don’t know whether it is frequently used in AST practices.

Lines 354-355: unclear by what is meant by “good representativeness of both *E. coli* and *Kp*”?

Reply: To clarify this, we have changed ‘good representativeness’ into ‘a sufficient number’ in lines 431-432 of the manuscript.

Line 356: Can the authors specify here what sequencing platforms/library preparations they used for NGS and TGS? They specify ONT sequencing in line 370 but don’t clarify the platform used to generate their short reads.

Reply: In lines 433-434, we referred to another article. In this article it is stated that isolates were subjected to NGS using the Illumina HiSeq 2500 (BaseClear). To clarify this, we also mentioned it in the manuscript in line 434: ‘In short, isolates were subjected to NGS using the Illumina HiSeq 2500 (BaseClear).’ The further details for ONT sequencing are described in the referred article.

Lines 356-369: perhaps re-order this paragraph to start with a description of NGS sequencing and library preparation before going into methodology involved in using the sequencing data

Reply: To limit the number of words we had decided to refer to a previous article for more details in lines 433-434. The information you request is described there.

Line 364: citations and/or web links missing for VirulenceFinder, SerotypeFinder and Kleborate

Reply: We have added references for these softwares in the method section. The website for Kleborate was already mentioned.

Lines 375-377: can the authors clarify why they focused on the *mcr-9* isolates and not say, the *mcr-1*?

Reply: We decided to focus on the *mcr-9* gene, since the *mcr-9* gene is more recently discovered and less is known about this gene. The *mcr-1* gene has already been characterised extensively.

Lines 379-284: Can the authors clarify why they used read mapping to ascertain mutations when they could have directly interrogated their assemblies for these gene sequences and then align? Does this yield any differences to what was observed using the read mapping approach?

Reply: We agree with the reviewer that the assemblies can be directly interrogated and we did do that as well. However, to make certain the observed mutations were not caused by assembly errors or by local low read coverage we always check by mapping the reads against the assembled gene sequences. All mutations identified by interrogating the assemblies were corroborated by mapping.

Figure 1: suggest adding in a breakdown of the 72 matched isolates by species i.e. n=X *E.coli* vs. *Kp*

Reply: As suggested, we added '(54 *E. coli* + 18 *K. pneumoniae*)' to figure 1.

Figure 3: what do the '6' and '61' in the figure denote?

Reply: To clarify this, we added '*For clusters, the allelic differences between the isolates are denoted.*' to the legend of figure 3.

Figure 4: suggest having the AMR genes and AMR class labels slanted on a 45 degree angle or so; currently difficult to read

Reply: If we slant the labels on a 45 degree angle, it is difficult to see to which antimicrobial resistance gene the columns belong and to which antibiotic class the genes belong. Furthermore, the figure would be too broad if we do that. In our opinion, this is the best and most compact manner to display this figure.

Minor comments:

Line 61: grammatical error: "had tested colistin-resistant"

Reply: As suggested, we changed 'was' into 'had' in line 100.

Line 169: typo "pneumoniae"

Reply: As suggested, we added 'e' in line 238.

Line 287: typo "than"

Reply: As suggested, we changed 'dan' into 'than' in line 363.

REVIEWERS' COMMENTS:

Reviewer #2 (Remarks to the Author):

I thank the authors for the time and effort in addressing each of the queries. I have no further comments.